# Nursing Diagnoses of Individuals with Myalgic Encephalomyelitis/Chronic Fatigue Syndrome: Research Protocol for a Qualitative Synthesis

**DOI:** 10.3390/healthcare10122506

**Published:** 2022-12-10

**Authors:** Cristina Oter-Quintana, Jesús Esteban-Hernández, Leticia Cuéllar-Pompa, María Candelas Gil-Carballo, Pedro Ruymán Brito-Brito, Angel Martín-García, María Teresa Alcolea-Cosín, Mercedes Martínez-Marcos, Almudena Alameda-Cuesta

**Affiliations:** 1Member of the Nursing and Health Care Research Group IDIPHISA, 28222 Majadahonda, Spain; 2Nursing Department, Faculty of Medicine, Autonomous University of Madrid, 28029 Madrid, Spain; 3Faculty of Health Sciences, Rey Juan Carlos University, 28922 Alcorcón, Spain; 4Medical Specialties and Public Health Department, Faculty of Health Sciences, Rey Juan Carlos University, 28922 Alcorcón, Spain; 5Care Research Institute, Illustrious Professional Association of Nurses of Santa Cruz de Tenerife, 30001 Santa Cruz de Tenerife, Spain; 6Library of Medicine, Faculty of Medicine, Autonomous University of Madrid, 28029 Madrid, Spain; 7Primary Care Management of Tenerife, The Canary Islands Health Service, 38204 La Laguna, Spain; 8Nursing Department, La Laguna University, 38200 La Laguna, Spain; 9Health Care Directorate (Southern District), Primary Care, Madrid Health Service, 28981 Parla, Spain; 10Nursing and Oral Medicine Department, Faculty of Health Sciences, Rey Juan Carlos University, 28922 Alcorcón, Spain

**Keywords:** fatigue syndrome, chronic, nursing diagnosis, qualitative research, nurses, systematic review

## Abstract

Although previously developed qualitative studies have explored the experience of illness of individuals with myalgic encephalomyelitis/chronic fatigue syndrome, these findings have not been undertaken for the purpose of enabling the identification of nursing care needs in such patients. This study aims to identify NANDA-I nursing diagnoses of adults with myalgic encephalomyelitis/chronic fatigue syndrome based on a qualitative literature review of their experience of illness. The protocol includes: searches in the electronic databases Medline, Embase, CINAHL, PsycINFO, SCI-EXPANDED, SSCI, SciELO, LILACS, and Cuiden; and manual searches in specialised journals and the references of the included studies. The authors will systematically search qualitative research studies published in databases from 1994 to 2021. Searches are limited to studies in Spanish and English. All stages of the review process will be carried out independently by two reviewers. Any disagreements shall be resolved through joint discussions, involving a third reviewer if necessary. The findings will be synthesised into a thematic analysis informed by the Domains and Classes of the NANDA-I Classification of Nursing Diagnoses, which will then serve to identify nursing diagnoses. This review will enable nursing professionals to identify the care needs of individuals with myalgic encephalomyelitis/chronic fatigue syndrome by taking into consideration their experience of illness in its entirety.

## 1. Introduction

Myalgic encephalomyelitis/chronic fatigue syndrome (ME/CFS) is a complex, multisystem, and chronic disease. The lack of consensus with regards to its definition makes it difficult to estimate its true prevalence, but a recent review places it at around 1% of the general population [1]. ME/CFS can affect individuals of any age and is most commonly diagnosed among women, with a male:female ratio of between 1:2 and 1:4, depending on the cohort analysed [2].

ME/CFS is characterised by a marked decline or impairment in the individual’s ability to perform academic, professional, social, and personal activities that persists for more than six months and is accompanied by intense fatigue that was not previously present and cannot be explained by excessive physical activity. Rest does not relieve the fatigue [3,4]. Post-exertional malaise is also one of the most characteristic manifestations of the disease and can appear immediately or within hours or days after physical and/or cognitive exertion, the intensity of which was previously well tolerated by the individual, lasting up to several weeks [4]. Other key symptoms include unrefreshing sleep, cognitive dysfunction, orthostatic intolerance, and pain (muscle pain, joint pain, and headache) [4]. Symptoms may remain or recur throughout the affected person’s life.

The prognosis of ME/CFS is uncertain, and it is estimated that the most of adult patients do not recover. Measurements of groups of individuals with ME/CFS show that their mean health-related quality of life is poorer than that of the general reference population and poorer than that of individuals who have suffered or are currently suffering from stroke, depression, or multiple sclerosis, among other conditions [5]. In the mildest form of ME/CFS, individuals are able to care for themselves and do minor household chores, although they may sometimes require assistance due to their limited mobility. They may continue to work and/or study, but at the cost of limiting their social life and leisure time, time they employ to recover and cope with the week. In the most severe cases of ME/CFS, individuals need assistance with basic care, are very sensitive to sensory stimuli, and remain at home in bed [6].

The aetiology of ME/CFS is not clearly defined. Immunological, neurological, endocrine, genetic, and psychiatric disorders and/or the presence of prior infections in affected individuals have been suggested as possible causes [7]. There are no specific biomarkers for the disease. In their absence, the diagnosis of ME/CFS is primarily clinical [4], with a variety of case definition criteria (e.g., The Revised Canadian Consensus Criteria, The International Consensus Criteria, The Institute of Medicine Criteria, etc.). These criteria are based on the presence of a varying number of symptoms and, to a lesser extent, on the severity and frequency of symptoms. The coexistence of ME/CFS with other conditions may delay its diagnosis or lead to misdiagnosis [3]. It is estimated that more than two-thirds of individuals living with ME/CFS take at least a year to be diagnosed, often after visiting several physicians [3].

There is no specific treatment for the disease [4]. The existing evidence on effective pharmacological and non-pharmacological interventions is limited and of poor or very poor quality. The therapeutic option involves, if necessary, devising a personalised treatment plan together with the individual and their caregivers [6] which focuses on the most problematic symptoms [8]. Energy management is one of the main recommended therapeutic strategies; it is geared towards planning activities to stay within the individual’s energy envelope, without forcing any activity, and resting as needed.

The lack of a clear aetiology, unequivocal diagnostic criteria, and specific treatments, as well as coexistence with other conditions, have contributed to questioning the ‘real’ nature of ME/CFS [8]. As a result, ME/CFS could be referred to as a ‘contested illness’, a term used by Swoboda to refer to illnesses whose ‘real’ existence is the subject of controversy and discussion by public authorities, healthcare professionals, and society in general [9]. The controversial nature of the disease has a devastating effect on affected individuals. Their journey through the healthcare system is fraught with frustration and suffering due to a lack of knowledge among healthcare professionals of the disease and their disbelief and scepticism about its ‘organic’ nature [10,11]. The symptoms of individuals with ME/CFS are trivialised and/or attributed to psychological issues [10]. Those affected experience difficulties accessing sick leave or other social benefits, despite the limitations they face because of their illness [12]. Attaining a diagnosis of ME/CFS is often linked to a long winding road [13] and arduous ‘negotiations’ with medical staff, which erodes trust, respect, and the therapeutic relationship [14]. As it is a contested nosological entity, the process of differentiation between the clinical label and the subjective experience is not finalised, leaving the individual, despite the diagnosis, in a sort of limbo that becomes a space of exclusion and vulnerability [15]. Affected people’s experiences of delegitimisation extend to their relationships with others [14,16]. The ‘invisibility’ of the illness, with individuals appearing healthy to those around them [7,16], and the fluctuation of symptoms, affecting in varying ways the ability of individuals to go about their day-to-day lives, contribute to increasing mistrust from partners and friends about its real nature. A lack of understanding from the people closest to them is an obstacle to requesting help in dealing with the physical limitations resulting from the disease, which can have negative consequences for their health [7]. At the same time, scepticism about the authenticity of their symptoms is experienced as a challenge to their own honesty and integrity, contributing to feelings of isolation and loneliness [14]. Moreover, ME/CFS, like any other chronic illness, is deeply embedded in the life paths of those affected, becoming a key element in the construction of one’s own subjectivity [15,17]. However, in the context of an illness marred by suspicion and stigma, the necessary identity transformations linked to chronicity will be more complicated for the individual, increasing their suffering and vulnerability.

Kleinman defined ‘experience’ on theoretical grounds as: “the intersubjective medium of social transactions in local moral worlds. It is the outcome of cultural categories and social structures interacting with psychophysiological processes such that a mediating world is constituted. Experience is the felt flow of that intersubjective medium” [18] (p.97). This conceptualisation is particularly interesting because of the emphasis placed on the relational aspects of the disease in everyday life. In individuals with ME/CFS, this notion of experience refers not only to the discomfort derived from the symptoms of the illness itself, but also to the suffering caused by the stigmatisation processes linked to its contested nature [19]. The notion of experience has been linked to nursing diagnoses, as stated in the definition approved at the ninth NANDA conference and subsequently modified in 2009. Defined as “a clinical judgment about individual, family, or community experiences/responses to actual or potential health problems/life process” [20] (p.515), nursing diagnoses provide the basis for the selection of nursing interventions that make it possible to achieve outcomes for which nurses are accountable [21]. Pivoting on the notion of experience/response, nursing diagnoses lead necessarily to the validation of people’s experiences of illness, while allowing the human condition to be present in healthcare. The NANDA-I taxonomy is a classification system organised into domains, classes, and diagnoses. Each domain represents a different sphere of knowledge in nursing. Classes group together nursing diagnoses that share a series of common attributes. The 2021–2023 edition includes 267 nursing diagnoses that represent human responses of individuals, families, or communities to health problems and/or life processes [21]. In the Spanish context, NANDA-I terminology is the linguistic diagnostic system used by professional nurses to enter information into the Electronic Health Record (EHR). This standardised language is especially useful in articulating the care needs of individuals with ME/CFS, given its potential for capturing groups of problems associated with the disease beyond a purely organic dimension.

Qualitative studies have the potential to provide an in-depth understanding of illness experiences, which is why the results of qualitative research have been used as empirical material for nursing diagnoses [22]. The qualitative findings of studies exploring the illness experience of individuals with ME/CFS may be considered as empirical material from which to extract and name the nursing diagnoses present in this population group. Although addressing symptoms is fundamental, providing care driven by nursing diagnoses, far from hindering clinical practice, makes it possible to capture (without fragmenting) discomforts that go beyond the physical manifestations of the illness and affect the very condition of the individual as such. It makes visible (and treatable) that space of experience that can be obscured in a healthcare system focused on the signs and, to a lesser extent, the suffering resulting from illness.

This study aims to identify NANDA-I nursing diagnoses of adults with ME/CFS based on a qualitative literature review of their experience of illness.

The questions for this review are as follows:What are the main themes emerging from studies on the experience of illness of adults with ME/CFS?What nursing diagnoses can be identified from the analysis of the experiences of illness of individuals with ME/CFS?

## 2. Materials and Methods

This study will explore the experiences and perspectives of individuals with ME/CFS. This review will include studies involving populations of women and men with ME/CFS aged 18 and above. Participants had to have been diagnosed with ME/CFS by a medical professional, irrespective of the diagnostic criteria used. Studies with a mixed population, i.e., involving individuals affected by ME/CFS together with those affected by other diseases, will not be included. Studies in which some or all the participants are children or adolescents will not be included.

This review will include research conducted in any geographical, cultural, or healthcare setting. Original qualitative research articles will be included in the review, regardless of their theoretical–methodological approach, as well as mixed studies, provided that their quantitative and qualitative components are clearly differentiated. The selection of qualitative studies is based on the consideration that this type of design allows us to gain an in-depth understanding of the experiences of individuals with ME/CFS. Quantitative studies, review articles, consensus documents, and grey literature will be excluded. As part of the planning process for this review, several manual searches in the PROSPERO and Trip databases were conducted to find studies already published on this topic. We will develop search strategies for each of the following online literature databases: Medline/Medline In-Process (OvidSP Interface), Embase (Embase Interface), CINAHL (EbscoHOST Interface), PsycINFO (EbscoHOST Interface), SCI-EXPANDED (WOS Interface), SSCI (WOS Interface), SciELO (WOS Interface), LILACS (Biblioteca Virtual de Salud España), and Cuiden. Initially, the search strategy will be tested by one author using the Embase database using different combinations of terms until the best possible balance between sensitivity and specificity of the results is achieved. Subsequently, the final search strategy will be adapted for use in the rest of the selected databases. The search will be limited to studies published in English and Spanish from 1994 to 2021. The limit of 1994 is due to the fact that it was in this year that the Fukuda criteria for the diagnosis of CFS were published, which are used widely around the world. Table 1 shows the search strategy designed for the Medline and Medline In-Process databases using the Ovid SP platform. The search will be conducted using subject headings (MeSH and Emtree terms) and keyword searches (limited to article title and abstract). References of included studies will also be reviewed to find other eligible studies. In addition, selected journals specialising in the topic will be searched manually. Eligible publications will be included in the selection process. If any of the results are selected, this will be counted as a manual search. The complete search strategy for each database, the number of hits retrieved, and the reasons for exclusion of any study at the full-text stage in the selection process will be recorded and provided as appendices. The last update search will be performed at the end of the analysis process to include the most recent literature.

A database will be created to include the references obtained in the literature search. One of the members of the research team will proceed to manually eliminate the duplicates. The titles and abstracts of the identified studies will be individually reviewed for eligibility by two members of the research team. This selection will be blinded as to authors and journal of publication of the manuscript to reduce selection bias based on these two aspects. Any disagreements will be resolved through joint discussions, involving a third member of the research team if necessary. A pilot test will be carried out beforehand with a sample of 10% of the gathered articles to ensure that there is agreement between reviewers in the application of the inclusion/exclusion criteria. The studies initially included will be independently reviewed in full text to reach agreement on the studies that will eventually be included in the review. Reference management software (Mendeley Deskstop version 1.19.8, Elsevier, London, UK ) will be used to assist in the management of references obtained throughout the selection process. If necessary, during the selection process, the first author or the corresponding author will be contacted to clarify the eligibility of their article for our review. The selection process will be carried out according to the Preferred Reporting Items for Systematic Review and Meta-Analysis Protocols (PRISMA-P guidelines) [23].

Two members of the research team will review the quality of the included studies independently. The Critical Appraisal Skills Program (CASP) checklist will be used for qualitative studies [24] and the Mixed Methods Appraisal Tool (MMAT) will be used for mixed studies [24,25]. To ensure agreement between reviewers in the application of the assessment criteria, a pilot test will be carried out beforehand with a sample of five articles. Any disagreements will be resolved through discussion, involving a third reviewer if necessary to achieve consensus. Following the criteria used by Sandelowski and Barroso [26], no study will be discarded at this stage, regardless of the score it obtains in the quality assessment. However, this score will be taken into account when assessing its relevance for the synthesis. A data extraction instrument has been developed based on the Joanna Briggs Institute’s approach for qualitative studies [27]. Information will be collected on the author, year and journal of publication, aims, theoretical–methodological approach, data setting, data collection method, sampling technique, and sociodemographic and clinical characteristics of the population. Findings will be extracted from the Results or Findings section of the abstract and text of the included studies. Following Sandelowsky’s proposal, the following will be considered as findings: “the data-driven and integrated discoveries, judgments, and/or pronouncements researchers offer about the phenomena, events, or cases under investigation” [28] (p.909). We will input the original articles into ATLAS.ti scientific software version 22, ATLAS.ti GmbH, Berlin, Germany. For qualitative data analysis to avoid any loss/error in the data extraction process, although only the findings contained in the articles will be analysed [29]. The data extraction tool will be pilot-tested with a sample of 10 randomly selected articles to resolve potential disagreements between reviewers. Two members of the research team will extract data from each study independently and subsequently consolidate their records. Any disagreements will be resolved by discussion, involving a third member of the research team if necessary.

The process of coding and synthesising the findings will be based on Thomas and Harden’s thematic synthesis approach [30]. The findings of the selected studies will be coded line by line and according to the domains and classes of the NANDA-I Classification, 2021–2023 edition [21]. Its domains and classes represent, at different levels of abstraction, the human responses of individuals to life processes/health problems attended to by nursing professionals. This coding strategy facilitates the subsequent process of identifying nursing diagnoses. To support the coding process, researchers will have at their disposal the domain and class definitions included in the 2021–2023 edition of the NANDA-I Classification [21]. Open coding will also be carried out to analyse any fragments that cannot be coded according to the existing domains and classes. Coding will be carried out by two members of the research team independently, who have prior experience with NANDA-I terminology. Once the findings have been organised according to domains and classes, the identification of NANDA-I nursing diagnoses will proceed. During the codification process, a specialised bibliography will be consulted if necessary regarding nursing diagnoses included in the classification to guarantee a deeper understanding of their underlying concepts. To this end, the researchers will compare the definitions and defining characteristics of the various NANDA-I nursing diagnoses with the findings of the selected qualitative studies [21]. For the sake of diagnostic accuracy, all potential diagnoses suggested by the data will be included.

Regarding the validity and reliability/rigour of the study, the different stages of the review process will be carried out by peers independently [29]. The study selection process will follow the Preferred Reporting Items for Systematic Review and Meta-Analysis Protocols (PRISMA-P) [23]. The qualitative synthesis report will be based on the Enhancing Transparency in Reporting the Synthesis of Qualitative Research (ENTREQ) statement [31]. A preliminary version of the protocol for this review has been registered in PROSPERO (CRD42020164196): https://www.crd.york.ac.uk/prospero/display_record.php?RecordID=164196 (accessed on 5 December 2022). The research team will hold regular meetings to discuss and agree on issues pertaining to the research process. These meetings will help to maintain the level of reflexivity necessary to ensure a rigorous review.

## 3. Discussion

This study aims to identify the nursing care needs of individuals with ME/CFS using NANDA-I nursing diagnoses. Their identification is not intended to ignore the necessary personalisation of the care plans to be developed in each case. This study aims to become primarily a guide informing, not determining, nursing care based on the empirical evidence provided by qualitative studies, thus fulfilling the requirement of basing nursing practice on scientific knowledge.

Although previously developed qualitative studies have explored the experience of illness of individuals with ME/CFS, synthesis of these findings has not been undertaken for the purpose of enabling the formulation of nursing diagnoses. The ‘translation’ of these findings into standardised NANDA-I nursing language is crucial, as it provides a ‘map’ of the care requirements of affected individuals that goes beyond addressing symptoms and takes the contested nature of the disease into consideration. On the basis of this ‘map’, clinically oriented care can be planned, which, a priori, has great potential to improve the quality of life and health of individuals living with ME/CFS.

## 4. Limitations

Caution should be exercised when interpreting the findings of this study due to its potential limitations. Firstly, it is important to highlight that only studies whose populations have been diagnosed with ME/CFS by a medical professional are to be included. Given the complexity of the diagnostic process itself and the controversy surrounding it, the use of this criterion will limit the studies to be included in this review. This will exclude findings of studies in which the study population is in the process of receiving a clinical diagnosis, studies in which it is not specified whether or not the diagnosis is available, and studies in which self-diagnosed individuals are included. However, the use of such a restrictive criterion ensures that the results of the review refer exclusively to individuals with ME/CFS. In addition, the use as empirical material of qualitative findings reported by original articles and, consequently, texts of limited length may compromise the accuracy of NANDA-I nursing diagnoses, which require specific information in line with the defining characteristics/risk factors associated with them. Thus, the research team will identify all the nursing diagnoses that are considered to result from reviewing the empirical evidence, which will, in turn, be reviewed by professional experts to ensure the reliability of the selected NANDA-I nursing diagnoses.

## 5. Conclusions

Progress in the identification of nursing diagnoses that take into consideration the experience of illness in its entirety of individuals with ME/CFS is essential to make visible (and treatable) the space of experience that can be obscured in a healthcare system focused on the signs of, and to a lesser extent, the suffering resulting from, illness.

Nursing professionals have a fundamental role to play in addressing the delegitimisation of ME/CFS and improving the quality of care provided to individuals with this disease. The identification of nursing diagnoses opens up the possibility of rethinking, if necessary, existing care models and spaces.

## Figures and Tables

**Table 1 healthcare-10-02506-t001:** The search strategy designed for the Medline and Medline In-Process databases using the Ovid SP platform.

Number	Search Strategy
1	exp Fatigue Syndrome, Chronic/
2	(‘chronic fatigue syndrome’ or ‘royal free disease’ or ‘systemic exertion intolerance disease’ or ‘myalgic encephalomyelitis’).ab,ti.
3	(‘chronic fatigue and immune dysfunction syndrome’).ab,ti.
4	(‘chronic fatigue’ adj2 (disorder or syndrome)).ab,ti.
5	(‘fatigue syndrome’ adj2 (chronic or postviral)).ab,ti.
6	OR/1–5
7	(‘emotional adaptation’ or ‘emotional adaptations’ or ‘psychologic adaptation’ or ‘psychological adaptation’ or ‘psychological adaptations’ or ‘adaptive behavior’ or ‘adaptive behaviors’).ab,ti.
8	(‘life change event’ or ‘event history analys?s’ or ‘life change events’).ab,ti.
9	((emotional or psychological) adj1 adjustments).ab,ti.
10	exp Attitude to Health/or *Emotional Adjustment/or *Life Change Events/or *Adaptation, Psychological/or exp Emotions/or exp Self Concept/or *Illness Behavior/or *Sick Role/or *Quality of Life/or *Social Stigma/or exp Social Support/or exp Interpersonal Relations/or *Family Relations/
11	(coping adj2 (behavior or behaviors or skill or skills)).ab,ti.
12	(emotions or emotion or regret or regrets or feeling or feelings).ab,ti.
13	(hrql or ‘health related quality of life’ or ‘quality of life’ or ‘life quality’).ab,ti.
14	(self adj1 (concept or concepts or perception or perceptions or confidence or esteem)).ab,ti.
15	((illness or sickness) adj1 (behavior or behaviors)).ab,ti.
16	(social adj1 (stigma or support)).ab,ti.
17	(‘interpersonal relation’ or ‘interpersonal relations’ or ‘interpersonal relationship’ or ‘social interaction’ or ‘social interactions’ or ‘social relationships’).ab,ti.
18	((husband or wife or partner) adj1 communication).ab,ti.
19	(gender adj1 (issues or issue or relation or relationship or relations)).ab,ti.
20	(family adj1 (dynamic or relation or relationship or dynamics or relations)).ab,ti.
21	((patient or illness) adj1 (experience or experiences)).ab,ti.
22	((lived or life) adj1 (experience or experiences)).ab,ti.
23	(‘health attitude’ or ‘health attitudes’).ab,ti.
24	OR/7–23
25	6 and 24
26	limit 25 to (yr = “1994—Current” and (English or Spanish))

* Strategy designed in Medline and Medline In-Process using the Ovid SP platform.

## Data Availability

Not applicable.

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
