# Peer review of "Nursing Diagnoses of Individuals with Myalgic Encephalomyelitis/Chronic Fatigue Syndrome: Research Protocol for a Qualitative Synthesis"

_healthcare, 2022, doi:10.3390/healthcare10122506_

Round 1

Reviewer 1 Report

This is an interesting paper proposing an original approach to identifying the care needs of people with ME/CFS.

Not all readers will be familiar with NANDA-I, and its possible usefulness in facilitating the identification of nursing care needs in ME/CFS. It would be helpful therefore if an additional paragraph could be included in the Introduction summarising this.

There are some minor problems in the text which should be addressed:-

Line 169: There is an incomplete sentence here, viz.: “It will be included studies who population …”

Line 187: The authors write about “… using different combinations of terms until the best possible balance between sensitivity and specificity of the result is achieved”. Could the authors please explain how sensitivity and specificity are to determined here, and indicate by what criteria the best possible balance is to be identified.

Lines 211-212: The authors state that: “A pilot test will be carried out beforehand with a sample of 100 articles …”. Could they please explain how the sample size of 100 was determined, and how they would cope with the situation if they were unable to identify this number of papers.

If these concerns could be addressed, the paper would be well worth publishing.

Author Response

Revisor 1

Not all readers will be familiar with NANDA-I, and its possible usefulness in facilitating the identification of nursing care needs in ME/CFS. It would be helpful therefore if an additional paragraph could be included in the Introduction summarising this.

Thank you for your comment. The following extract has been added:

(Line 147) The NANDA-I taxonomy is a classification system that allows for sorting into three areas of interest for professional nurses. It is organised into domains, classes, and diagnoses. Each domain represents a different sphere of knowledge in nursing. Classes group together nursing diagnoses that share a series of common attributes. The 2021-2023 edition includes 267 nursing diagnoses that represent human responses of individuals, families or communities to health problems and/or life processes [21]. In the Spanish context, NANDA-I terminology is the linguistic diagnostic system used by professional nurses to enter information into the electronic health record (EHR). This standardised language is especially useful in articulating care needs of individuals with EM/SFC given its potential for capturing groups of problems associated with the disease, beyond a purely organic dimension.

Line 169: There is an incomplete sentence here, viz.: “It will be included studies who population …”.

 Thank you for your feedback. The highlighted sentence has been removed, as the study population was previously described. 

Line 187: The authors write about “… using different combinations of terms until the best possible balance between sensitivity and specificity of the result is achieved”. Could the authors please explain how sensitivity and specificity are to determined here, and indicate by what criteria the best possible balance is to be identified.

Thank you very much for your suggestions. Within the methodology, this segment makes reference to a key step within the design process of an optimal research strategy. When a systematic search is undertaken, there is always doubt regarding whether the research has managed to collect all or the majority of potentially relevant articles. This is difficult to know because it is not widely known which, nor how many, articles are relevant. For this reason, it is advisable to develop several initial tests using a wide range of strategies, primarily so that the literature review is more sensitive, but then also to verify whether new relevant articles are found using more specific literature reviews, the results of which can be compared with those of the initial search [1]. For this, the summaries are examined in order to determine the potential relevance of their results. As recommended by The Cochrane Information Retrieval Methods Group in the Cochrane Handbook for Systematic Reviews of Interventions, this procedure allows us to find a balance between sensitivity and precision. The following table details the formula used to calculate the sensitivity and the precision of a search [2]. However, for our study, seeing as our objective is not to validate the search strategy, our intention is to estimate how balanced these two results are using the process previously described, and in consensus with the authors in charge of reviewing the search strategies and results.

Reports retrieved

Reports not retrieved

Relevant reports

Relevant reports retrieved (a)

Relevant reports not retrieved (b)

Irrelevant reports

Irrelevant reports retrieved (c)

Irrelevant reports not retrieved (d)

Sensitivity: fraction of relevant reports retrieved from all relevant reports (a/(a+b))

Precision: fraction of relevant reports retrieved from all reports retrieved (a/(a+c))

[1] Bramer WM, de Jonge GB, Rethlefsen ML, Mast F, Kleijnen J. A systematic approach to searching: an efficient and complete method to develop literature searches. J Med Libr Assoc. 2018; 106(4):531-541. doi: 10.5195/jmla.2018.283.

[2] Lefebvre C, Glanville J, Briscoe S, Featherstone R, Littlewood A, Marshall C, Metzendorf M-I, Noel-Storr A, Paynter R, Rader T, Thomas J, Wieland LS. Chapter 4: Searching for and selecting studies. In: Higgins JPT, Thomas J, Chandler J, Cumpston M, Li T, Page MJ, Welch VA (editors). Cochrane Handbook for Systematic Reviews of Interventions version 6.3 (updated February 2022). Cochrane, 2022. Available from www.training.cochrane.org/handbook.

Lines 211-212: The authors state that: “A pilot test will be carried out beforehand with a sample of 100 articles …”. Could they please explain how the sample size of 100 was determined, and how they would cope with the situation if they were unable to identify this number of papers.

 Thank you very much for your comments. There were various search attempts of several databases. Given the provisional results that were obtained, it was deemed sufficient to carry out the initial pilot test with 10% of the total number of gathered articles. The number 100 was calculated from this percentage. It has been modified in-text, remarking that the pilot test will be carried out with 10% of the total gathered studies (Line 228)

Reviewer 2 Report

The study protocol is well thought out-bringing in double reviewers and using established criteria to review and judge the strength of the work.

I would suggest adding in something so that reviewers have expertise in NANDA as there are currently around 265 diagnoses. Granted the authors have hypothesize in the search criteria what some of the diagnoses may be-but I worry there is a need to not bias or narrow the work too early as something new may come to light.

I am curious why you submitted the proposed protocol and did not conduct it first-and share the results in an article. That would be more of a substantial contribution to the literature.

Author Response

I would suggest adding in something so that reviewers have expertise in NANDA as there are currently around 265 diagnoses. Granted the authors have hypothesize in the search criteria what some of the diagnoses may be-but I worry there is a need to not bias or narrow the work too early as something new may come to light.

 Thank you very much for your suggestions. We have included in the text that coding will be carried out by two members of the research team independently, who have prior experience with NANDA-I terminology, and that, during the codification process, a specialised bibliography will be consulted if necessary regarding nursing diagnoses included in the classification to guarantee a deeper understanding of their underlying concepts. In response to the reviewer’s concerns, in the case of the NANDA-I terminology, an open coding has been chosen for those text fragments that cannot be coded using the current components of the NANDA-I terminology. (Line 274-279)

 I am curious why you submitted the proposed protocol and did not conduct it first-and share the results in an article. That would be more of a substantial contribution to the literature.

Thank you very much for your comment. We find it pertinent to publish the revision protocol as a way to guarantee the investigative study that will be undertaken at a later stage. We also understand that, if considered fit for publishing, it may be useful in carrying out other similar investigation protocols. At a later stage, and following the recommendations of the reviser, we are indeed, very interested in publishing the results of this work.

Reviewer 3 Report

Dear authors,

I am very happy to be a part of this review, you have made a great and scientifically sound manuscript detailing some of the key aspects of such a debilitating yet under (properly) researched disease. Thank you for this. I have little to amend or correct, here are a few points:

- line 102 etc you talk about the frustrating journey through the healthcare system and with healthcare professionals and the consequences of this for patients with ME/CFS, I would also mention the problems patients with ME/CFS have in getting medical retirement (adequate payment) because of it being a ‘contested illness’.

- The word ”feel” in line 104, I understand that you probably used this to highlight how patients experience this but I find this word quite subjective and can almost mislead to people thinking this is not actually true but just a perception, so I would consider changing the word “feel”, instead you could word the sentence like this: “The symptoms of individuals with ME/CFS are trivialised and/or attributed to psychological issues [10]” 

- in line 155 you mention NANDA for the first time, for a random reader it would be helpful if you explained what NANDA is the first time you mention it (even though very briefly mentioned in the abstract)

- line 169 “It will be included studies who population” this is not a full sentence, what are you trying to say?

-in line 190 you mention that you will be limiting your search to studies published after 1994. Why did you select this cutoff date?

I wish you lots of success with your article.

kind regards,

Author Response

- line 102 etc you talk about the frustrating journey through the healthcare system and with healthcare professionals and the consequences of this for patients with ME/CFS, I would also mention the problems patients with ME/CFS have in getting medical retirement (adequate payment) because of it being a ‘contested illness’.

 Thank you very much for your comment. We have added the following sentence: Those affected experience difficulties accessing sick leave or other social benefits, de-spite the limitations they face because of their illness. (Line 106-108)

- The word ”feel” in line 104, I understand that you probably used this to highlight how patients experience this but I find this word quite subjective and can almost mislead to people thinking this is not actually true but just a perception, so I would consider changing the word “feel”, instead you could word the sentence like this: “The symptoms of individuals with ME/CFS are trivialised and/or attributed to psychological issues [10]” 

 Thank you for your comment. The sentence has been amended as indicated by the reviewer.

- in line 155 you mention NANDA for the first time, for a random reader it would be helpful if you explained what NANDA is the first time you mention it (even though very briefly mentioned in the abstract)

Thank you for your comment. A paragraph has been added to this effect.(Line 147)

- line 169 “It will be included studies who population” this is not a full sentence, what are you trying to say?

 Thank you for your comment. The highlighted sentence has been deleted.

-in line 190 you mention that you will be limiting your search to studies published after 1994. Why did you select this cutoff date?

Thank you for your comment. This date was specified as it was in this year that the Fukuda criteria were published for diagnosing CFS, and they represent an important milestone in the recognition of the disease. These criteria continue to be widely used on a global scale.